# Effect of Fumonisin B1 on Proliferation and Apoptosis of Intestinal Porcine Epithelial Cells

**DOI:** 10.3390/toxins14070471

**Published:** 2022-07-09

**Authors:** Tianjie Wang, Hongyu Lei, Lihua Zhou, Meiwen Tang, Qing Liu, Feng Long, Qing Li, Jianming Su

**Affiliations:** Provincial Key Laboratory of Protein Engineering in Animal Vaccines College of Veterinary Medicin, Hunan Agricultural University, Changsha 410128, China; tjwang@stu.hunau.edu.cn (T.W.); leihy77@hunau.edu.cn (H.L.); zlh240141@163.com (L.Z.); 1737151498@stu.hunau.edu.cn (M.T.); liuqingqing@stu.hunau.edu.cn (Q.L.); 1499685240@stu.hunau.edu.cn (F.L.); qingli@stu.hunau.edu.cn (Q.L.)

**Keywords:** Fumonisin B_1_, intestinal porcine epithelial cells, apoptosis, cell proliferation

## Abstract

Fumonisin B_1_ (FB_1_), which is a mycotoxin produced by *Fusarium moniliforme* and *Fusarium rotarum*, has a number of toxic effects in animals. Moldy feed containing FB_1_ can damage the intestine. In this study, we used intestinal porcine epithelial cells (IPEC-J2) as an in vitro model to explore the effects of FB_1_ on cell cycle and apoptosis. The results showed that IPEC-J2 cells treated with 10, 20, and 40 μg/mL FB_1_ for 48 h experienced different degrees of damage manifested as decreases in cell number and viability, as well as cell shrinkage and floating. In addition, FB_1_ reduced cell proliferation and the mRNA and protein expression of proliferating cell nuclear antigen (PCNA), cyclin-dependent kinase 2 (CDK2), CDK4, cyclinD1, and cyclinE1. FB_1_ blocked the cell cycle in the G1 phase. FB_1_ also induced mitochondrial pathway apoptosis, reduced mitochondrial membrane potential, and promoted mRNA and protein expression of Caspase3, Caspase9, and Bax. The findings suggest that FB_1_ can induce IPEC-J2 cell damage, block the cell cycle, and promote cell apoptosis.

## 1. Introduction

Mycotoxins produced by fungi are considered as contaminants in animal feed [1]. Fumonisin B_1_ (FB_1_) is produced by Fusarium verticillioides, Fusarium proliferatum, and other Fusarium species; it is often found in corn [2]. FB_1_ has been classified as a group 2B hazard by international research institutions [3] due to its global distribution and its harmful effects in humans and animals [4]. The gastrointestinal system is the first barrier against ingested toxins [5]. Following ingestion of FB_1_-contaminated food or feed, the intestinal epithelium will be continuously exposed to high levels of FB1 [6]. Intestinal exposure to FB1 can lead to villous fusion and atrophy in the intestines of pigs [5]. In addition, FB1 impairs the establishment of the epithelial barrier and disrupts epithelium that is already established [7]. These injuries not only cause intestinal dysfunction but also promote FB_1_ entering the circulatory system. As FB_1_ spreads via the circulatory system throughout the body, it can damage other organs, leading to a variety of diseases.

Cell proliferation is one of the basic life activities involved in homeostasis, and the process proceeds through a sequence of stages known as the cell cycle [8]. The cell cycle is divided into a synthesis phase (S), two intervening gap phases (G1 and G2), and a mitotic segregation phase (M) [9]. Whether the cell cycle progresses smoothly has important effects on cell proliferation and apoptosis. A variety of proteins are involved in cell proliferation and regulation of the cell cycle [10], including proliferating cell nuclear antigen (PCNA), cyclin-dependent kinases (CDKs), and CDK inhibitor (CKI) [11,12]. Cyclin D can interact with CDK4 and CDK6 to form a cyclin D-CDK4/6 complex. The cyclin D-CDK4/6 complex regulates the cell cycle by targeting pocket proteins (RB, P107, P130) [13]. Many foreign substances such as bacteria, viruses, and toxins may affect cell proliferation and cycle progression. As a result, FB1 is likely to affect the activation and interaction of cycle-related proteins, further arresting the cell cycle [14].

Apoptosis, which is a form of programmed cell death, functions to maintain systemic balance. Morphological hallmarks of apoptosis include cell shrinkage, membrane blebbing, nuclear condensation, and DNA fragmentation [15]. Classical biochemical markers of apoptosis include activation of proteases, caspases, and mitochondrial outer membrane permeabilization (MOMP). The Bcl-2 family, Apaf-1, and the caspase family in mammals are major components of the mitochondrial apoptosis pathway [16]. In the mitochondrial pathway, the critical event is a change in permeability of the outer mitochondrial membrane (OMM) resulting in the release of cytochrome C into the cytoplasm. When proteins released from mitochondria enter the cytoplasm, they activate apoptosis-related genes such as Caspase9 and Caspase3 and thus induce apoptosis [17].

Recent studies by us have shown that FB1 interferes with the expression of nutrient transporter genes in intestinal porcine epithelial cells (IPEC-J2) [18]. Moreover, FB_1_ inhibits the expression of tight-junction proteins by destroying the barriers [19]. In this study, we will explore the effects of FB_1_ on cell proliferation and apoptosis, and whether FB_1_ affects cell apoptosis through the mitochondrial pathway.

## 2. Results

### 2.1. FB_1_ Reduced the Viability of IPEC-J2 Cells and Caused Cell Damage

Cell survival was measured 24 and 48 h after IPEC-J2 cells were exposed to FB_1_ (Figure 1a). Cells exposed to FB_1_ at 10, 20, and 40 μg/mL for 24 h showed no significant changes in cell viability. After exposure to FB_1_ at 20 and 40 μg/mL for 48 h, cell viability decreased to 95.14% and 83.66%, respectively. Cell morphology was observed under a microscope after 24 h exposure to FB_1_ (Figure 1b,c). As the concentration of FB_1_ increased, the number of cells gradually decreased; the cells shrank, becoming rounded and floating. This suggests that FB_1_ could reduce cell survival and cause cell damage.

### 2.2. FB_1_ Decreased Cell Proliferation and Arrested the Cell Cycle

After 48 h of treatment with different concentrations of FB_1_, the fluorescence intensity of the 20 and 40 μg/mL FB_1_ treatment groups decreased continuously (Figure 2a), especially in the 40 μg/mL FB_1_ group. This indicated that the proliferation of IPEC-J2 cells was significantly inhibited with the increase in FB_1_ concentration after treatment for 48 h.

To further verify the effect of FB1 on cell proliferation, mRNA expression levels of PCNA, cyclin-dependent kinase inhibitors P21 and P27, cyclin-dependent kinases 2 and 4, cyclin D1 and cyclin E1 were detected (Figure 2b,c). The mRNA expression levels of PCNA and CyclinE1 decreased in a concentration-dependent manner. CDK2 expression decreased significantly under different concentrations of FB_1_. CDK4, and CyclinD1 mRNA expression levels decreased significantly after the 40 μg/mL FB_1_ treatment. The mRNA expression levels of P21 and P27 slightly increased in the 40 μg/mL FB_1_ group and the 20 μg/mL FB_1_ group. Meanwhile, the protein expression of CDK2 was significantly decreased in 20 μg/mL group, the expression of CDK4 acyclinD1 was significantly decreased in 40 μg/mL group (Figure 2d). Flow cytometry results showed that after FB_1_ treatment, the percentage of G1 phase cells increased from 65.51% to 83.55%, while the percentage of S phase cells decreased from 18.88% to 7.08%, and there was no significant change in the number of G2 phase cells. (Figure 2e) These results indicated that FB_1_ could inhibit the expression of G1/S phase-related genes, promote the mRNA expression of the cycle suppressor genes P21 and P27, block the cell cycle in G1 phase, and thus inhibit cell proliferation.

### 2.3. FB_1_ Reduced Mitochondrial Membrane Potential and Promoted Cell Apoptosis

Mitochondria are the main production sites of cell metabolism, and their function is important for maintaining normal cell development. JC-1 was used to reflect the changes in mitochondrial membrane potential (Figure 3a). The results showed that the red fluorescence intensity increased in the 10 μg/mL FB_1_ group but decreased in the 20 μg/mL and 40 μg/mL FB_1_ groups. Compared with the control group, the intensity of green fluorescence increased gradually, being extremely significant in the 20 μg/mL FB_1_ group and the 40 μg/mL FB_1_ group, suggesting that FB_1_ treatment could cause mitochondrial damage and loss of mitochondrial membrane potential in IPEC-J2 cells.

qPCR and Western blots were used to examine mRNA transcription levels and protein expression levels of pro-apoptotic genes *Caspase3*, *Caspase9*, *Bax* and the anti-apoptotic gene *Bcl-2* (Figure 3b). With the increase in FB_1_ concentration, mRNA and protein expression levels of *Caspase3*, *Caspas9*, and *Bax* gradually increased, and the mRNA transcription levels reached a peak at 20 μg/mL. The mRNA expression of *Bcl-2* decreased with the increase in FB_1_ concentration. Meanwhile, in terms of protein expression, the expression levels of cleaved Caspase9 and Bax increased significantly in the 20 μg/mL group, while the concentration of cleaved Caspase3 reached a peak at 40 μg/mL (Figure 3c). Flow cytometry showed that the proportions of early and late apoptotic cells increased in a concentration-dependent manner (Figure 3d,e). In particular, the proportion of apoptotic cells increased from 1.89% to 23.7%, while the proportion of living cells decreased from 96.5% to 70.7%. These results suggested that FB_1_ could promote the apoptosis of IPEC-J2 cells through the endogenous apoptosis pathway.

## 3. Discussion

As a common mycotoxin, FB_1_ is widely distributed in moldy grain and feed [20]. After the animal continues to ingest contaminated feed, FB_1_ enters the body through the digestive tract. Thus, an intact digestive tract helps prevent toxins from entering the circulatory system and reduces the toxic effects of FB_1_ [21]. IPEC-J2 cells are an in vitro model of the porcine intestine that can reflect the intestinal epithelial state when stimulated by FB_1_ and provide insights into the effects of FB_1_ on the porcine intestine.

The MTT results showed that when the FB_1_ treatment time was extended to 48 h, cell survival rate decreased in a concentration-dependent manner. The cell viability of the 10 μg/mL treatment group increased, a result that may be due to the short-term stress on the cells. In our previous study, 50 μg/mL FB_1_ also significantly reduced IPEC-J2 cell activity [19]. The morphology of cells can also reflect the degree of damage to cells. HE staining can enable us to observe the morphological changes in cells more clearly.

Cell cycle regulation is vital to cell proliferation, growth, and repair. The cell cycle is regulated by a variety of enzymes, proteins, cytokines, and cycle-related proteins [22]. CyclinD1 is an important protein in the G1 phase to S phase transition. It is expressed in early G1 phase and can bind to CDK4 or CDK6 to activate CDK2, enabling cells to pass the G1/S control point [23,24]. Anti-proliferative proteins such as P21 and P27 can inhibit the release of E2F by interacting with the cyclin/CDK complex [25]. Therefore, treatment of IPEC-J2 with FB_1_ can reduce the expression of CDK2, CDK4, CyclinD1, and CyclinE1, preventing cells from passing through the G1/S control point and arresting the cell cycle in the G1 phase, thus inhibiting cell proliferation. In addition to IPEC-J2, FB_1_ can inhibit the proliferation of a variety of cells. For example, pig spleen cells treated with FB_1_ displayed IC_50_ at low concentrations, but FB_1_ was less toxic to pig and human peripheral blood mononuclear cells (PBMCs), and PBMC proliferation was inhibited by 50% only at higher concentrations of the toxin [26]. Our previous studies have shown that FB_1_ inhibits SUVEC proliferation and prevents cells from entering the S phase from the G1 phase [15]. A variety of mycotoxins can affect cell proliferation, including AFB1 that can inhibit cell proliferation in porcine early embryonic development [27]. Similarly, 2000 mg/mL concentration of the Fusarium-derived mycotoxin deoxynivalenol (DON) significantly reduced the viability of IPEC-1 and IPEC-J2 cells and increased the proportion of cells in the G2/M phase [28].

Apoptosis is an evolutionarily conserved form of programmed cell death that is critical to animal development and tissue homeostasis [29]. Apoptosis includes type I apoptosis mediated by death receptors and type II apoptosis dependent on mitochondria [30]. In type II apoptosis, when mitochondria receive external stimuli, the membrane permeability of the mitochondria will change, and cytochrome C in mitochondria will be released into the cytoplasm to activate downstream apoptosis-related proteins. The expression of Caspase3 is one of the markers of apoptosis. Caspase3 is the upstream promoter of apoptosis. After cells receive apoptotic signals, Caspase3 activates the apoptotic executor Caspase9, thereby hydrolyzing a series of proteins and promoting apoptosis [31]. As a mycotoxin, FB_1_ can produce toxic effects on a variety of cells and promote cell apoptosis. In our study, FB_1_ significantly promoted the cleavage and expression of Caspase3 and Caspase9. Similarly, FB_1_ induced apoptosis of HK-2 cells in a dose-dependent manner [32]. FB_1_ at 50 μM (36 μg/mL) induced a large amount of DNA damage and chromatin depolymerization in SH-SY5y neuroblastoma cells, and also promoted nuclear translocation of apoptosis-inducing factors [33]. In terms of toxicity to animals, a significant increase in hepatocyte apoptosis was detected in mice fed with 5 mg/kg FB_1_ for 42 days [34]. In conclusion, FB_1_ displays cytotoxicity to a variety of animal cells and is capable of inducing apoptosis.

## 4. Conclusions

Our study demonstrated that FB_1_ can cause injury to IPEC-J2 cells. The main manifestations of cell damage were cell number reduction, cell shrinkage, and floating. FB_1_ can block the cell cycle and inhibit cell proliferation by reducing the expression of genes and proteins associated with the cell cycle (Figure 4). Meanwhile, FB_1_ can induce apoptosis through the mitochondrial apoptosis pathway.

## 5. Materials and Methods

### 5.1. Cells and Cell Culture

IPEC-J2 cells were bought from GuangZhou Jennio Biotech Co., Ltd. (Guangzhou, China). The cells were grown in DMEM (10% FBS + 1% penicillin/streptomycin). IPEC-J2 cells were cultured in an incubator at 37 °C with a continual supply of 5% CO_2_.

### 5.2. Cell Viability Assay

Cell viability was determined using the 3-(4,5-dimethylthiazol-2yl)-2,5-diphenyltetrazolium bromide assay (MTT, CWBIO, Beijing, China). FB_1_ (purity ≥ 98%, Sigma-Aldrich, St. Louis, MO, USA) was dissolved with dimethyl sulfoxide (DMSO, CWBIO) and its concentration was adjusted to 20 mg/mL. IPEC-J2 cells were seeded in cell culture plate and stimulated with different concentrations of FB_1_ (0, 10, 20, and 40 μg/mL), and samples were incubated for 24 and 48 h. After the treatments, cells were washed with DMEM and then incubated in MTT solution (CWBIO, Beijing, China) for 2 h. The supernatant was removed after incubating for 2 h, and DMSO was added. Finally, the absorbance of the solution was measured at 490 nm in a spectrophotometer after 10 min.

### 5.3. Observation of Cell Morphology

An IPEC-J2 cell suspension was dropped into a 24-well plate and covered with a cell culture coverslip. Different concentrations of FB_1_ (0, 10, 20, and 40 μg/mL) were added when the cell density reached 60%, and the IPEC-J2 cells were added with FB_1_ and cultured for 48 h. The cell culture coverslip was taken for fixation, dehydration, transparency, hematoxylin, and eosin staining. The cells were observed under a light microscope.

### 5.4. Cell Proliferation Assay

For EdU labeling, an IPEC-J2 cell suspension was dropped into a 24-well plate. Different concentrations of FB1 (0, 10, 20, and 40 μg/mL) were added to the 24-well plate, and the cells were cultured for 48 h. IPEC cells were treated with 10 μM EdU for 2 h. For EdU staining, the cells were fixed with 4% paraformaldehyde, washed three times with PBS, and then incubated in 0.3% Triton^®^ X-100(Beijing Solarbio Science and Technology Co., Ltd. Beijing, China) in PBS for 15 min at room temperature. The cells were then incubated with freshly-made Click-iT reaction cocktail for 15 min. Finally, images are taken under a fluorescence microscope.

### 5.5. RT-qPCR Analysis

Total RNA was isolated from IPEC-J2 cells by using TransZol Up reagent (TransGen Biotech, Beijing, China). Then, cDNA was synthesized using a reverse transcription kit according to manufacturer’s instructions (TransGen Biotech). Real-time PCR was performed using Taq Pro Universal SYBR qPCR Master Mix (Vazyme Biotech Co., Ltd. Nanjing, China) and oligonucleotide primers (Table 1) on a Real-Time PCR Detection System (Applied Biosystems, Thermo Fisher Scientific, Waltham, MA, USA), and gene expression was calculated by the 2^−∆∆Ct^ method.

### 5.6. Western Blot

Cells were treated with the above concentrations of FB_1_ for 48 h, then washed three times with ice-cold PBS. The cells were then lysed with RIPA buffer (Solarbio) containing a protease inhibitor mixture (Solarbio, Beijing, China) and protein phosphatase inhibitor (Solarbio). The total protein concentrations were determined using a Bradford protein kit (Tiandz, Beijing, China). The total proteins were resolved by electrophoresis and transferred to poly(vinylidene fluoride) membranes for 80 min at 200 mA. PVDF membranes were incubated with primary antibodies caspase3 (Beijing Biosynthesis Biotechnology Co., Ltd. Beijing, China), caspase9 (Bioss), CDK2 (Bioss), CDK4 (Bioss, Beijing, China), cyclinD1 (immunoway, Plano, TX 75024, USA), cyclinE1 (immunoway), Bax (Proteintech Group, Inc., Wuhan, China), and GAPDH (Servicebio, Wuhan, China) overnight at 4 °C. Then, the membrane was washed three times with Tris-buffered saline tween (TBST) for 15 min. Secondary antibody (IgG, Vazyme) was used to incubate the membranes for 1 h at 4 °C. Finally, band density was quantified using the Image Lab 5.0 on the ChemiDOC XRS + system (Bio-Rad, Hercules, CA, USA).

### 5.7. Cell Cycle Assay

After 48 h of treatment with FB_1_ (0, 10, 20, and 40 µg/mL), IPEC-J2 cells were suspended in ice-cold PBS after centrifugation. Then, 500 µL of propidium iodide (PI) solution (Genview, Beijing, China) was added to the sample and incubated for 30 min. Cells were analyzed using a flow cytometer.

### 5.8. Cell Apoptosis Assay

Apoptosis of IPEC-J2 cells was analyzed using a flow cytometer. Briefly, cells were treated with the above concentrations of FB_1_ for 48 h, then washed three times with ice-cold PBS. The cells were stained with Annexin V/FITC-propidium iodide (PI), and the sample was incubated in the dark for 30 min. Cells were analyzed on a flow cytometer.

### 5.9. ∆ψm Determination

IPEC-J2 were treated with different concentrations of FB_1_ for 48 h. ∆ψm of IPEC-J2 was observed by an inverted fluorescence microscope with a ∆ψm assay kit (Beyotime Biotech, Shanghai, China) according to the product’s protocol.

### 5.10. Statistical Analysis

Results were expressed as mean ± standard deviation (SD). Comparisons between groups were made using *t*-test (two groups) and one-way analysis of variance (ANOV A). All experiments were repeated at least three times. A value of *p* < 0.05 was considered as a statistically significant difference (data marked with an asterisk *).

## Figures and Tables

**Figure 1 toxins-14-00471-f001:**
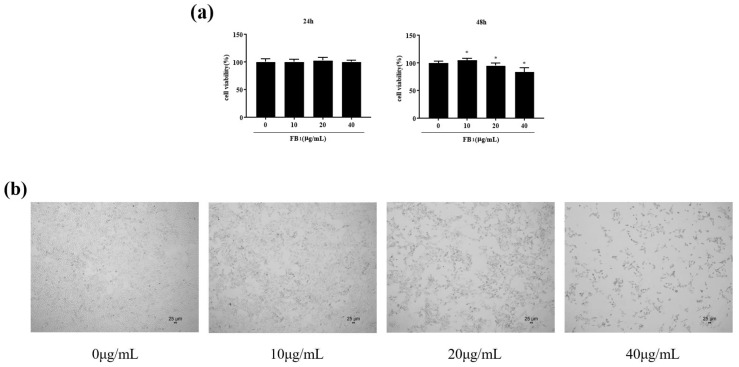
FB_1_ reduced cell viability. (**a**) The changes in cell viability after 24 and 48 h exposure to FB_1_ were detected (40×). (**b**) The morphological changes of cells treated with FB_1_ (0, 10, 20, and 40 μg/mL) for 48 h were observed under a microscope. (**c**) Cell morphology was observed by HE staining (100×). The data are expressed as mean ± SD. * *p* < 0.05.

**Figure 2 toxins-14-00471-f002:**
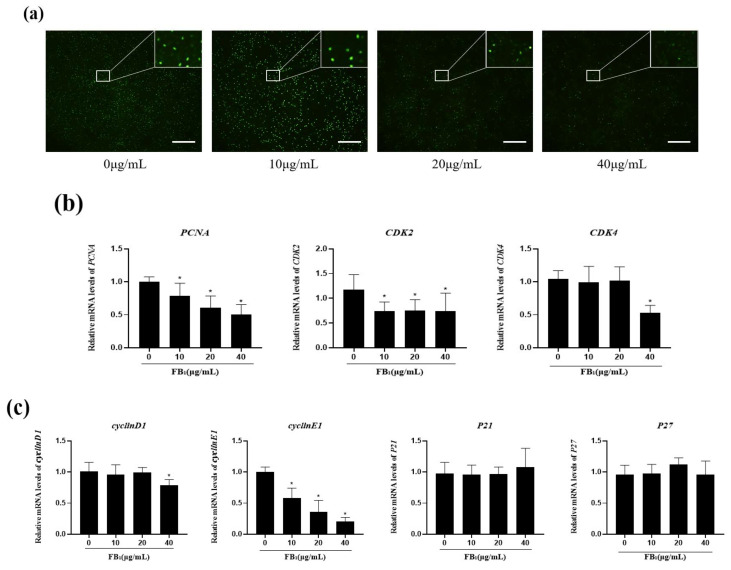
FB_1_ inhibits cell proliferation by blocking the cell cycle. (**a**) EdU analysis of the effect of FB_1_ on cell proliferation (40×). Scale bars: 50 μm. (**b**) The effects of FB_1_ on *PCNA*, *CDK2*, and *CDK4* mRNA were detected by qPCR. (**c**) The effects of FB_1_ on *cyclinD1*, *cyclinE1*, *P21*, and *P27* mRNA were detected by qPCR. (**d**) The effects of FB_1_ on cleaved CDK2, CDK4, cyclinD1 and cyclinE1 protein expression were measured. (**e**) Flow detection diagram of the cell cycle of IPEC-J2 treated with FB_1_ for 48 h. The data are expressed as mean ± SD. * *p* < 0.05.

**Figure 3 toxins-14-00471-f003:**
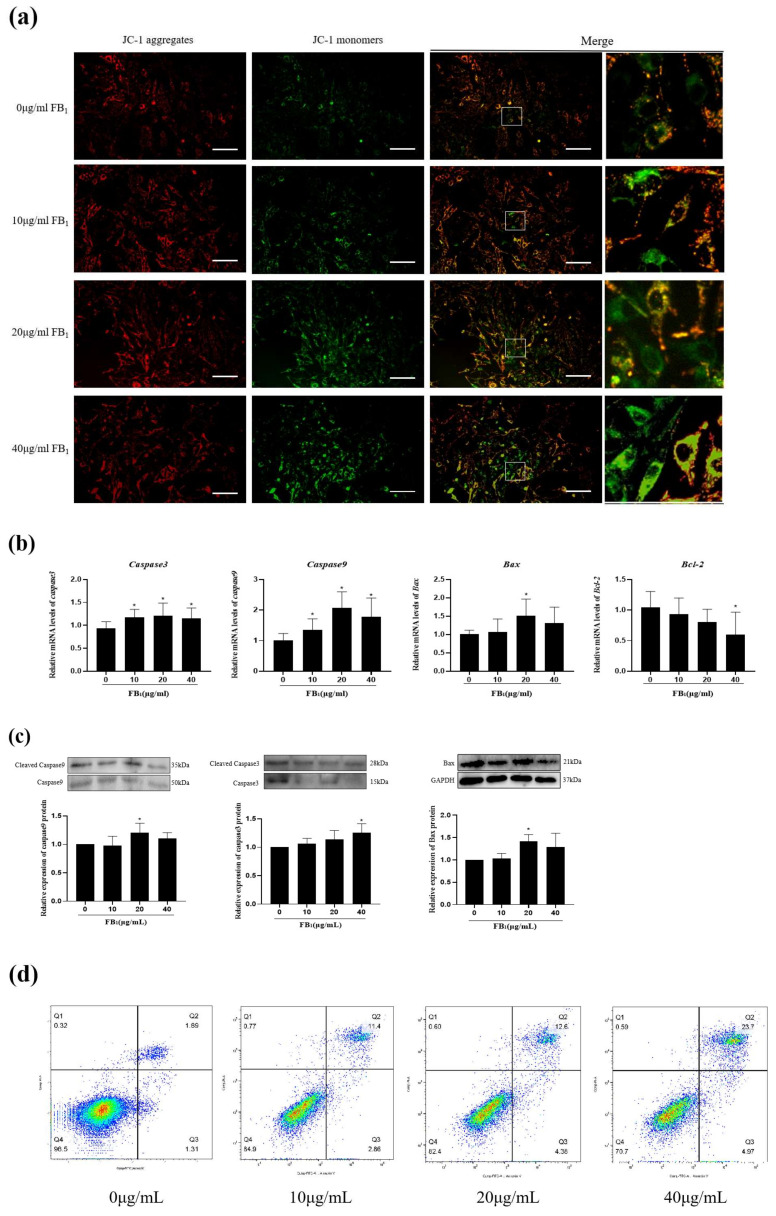
FB_1_ reduced ∆ψm and promoted cell apoptosis. (**a**) The effects of different concentrations of FB_1_ on mitochondrial membrane potential were detected by JC-1 (200×). Scale bars: 50 μm. (**b**) The effects of FB_1_ on *Caspase3*, *Caspase9*, *Bax*, and *Bcl-2* mRNA were detected by qPCR. (**c**) The effects of FB_1_ on cleaved Caspase9, cleaved Caspase3, and Bax protein expression were measured. (**d**) The apoptosis of IPEC-J2 cells after FB_1_ treatment was analyzed by flow cytometry. (**e**) The proportion changes of IPEC-J2 normal cells, early apoptotic cells, late apoptotic cells, and necrotic cells after FB_1_ treatment were measured by flow cytometry. The data are expressed as mean ± SD. * *p* < 0.05.

**Figure 4 toxins-14-00471-f004:**
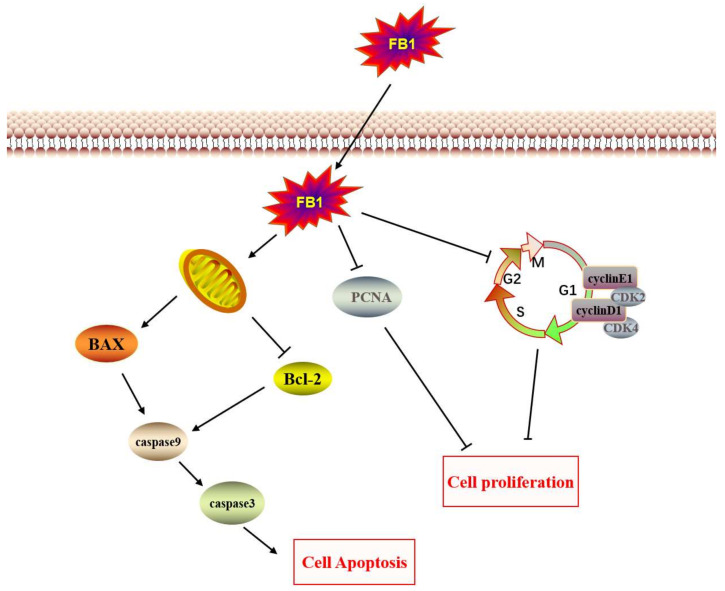
The supposed mechanism of toxic action of FB1 on IPEC-J2 cells.

**Table 1 toxins-14-00471-t001:** Oligonucleotide primers used for real-time PCR.

Primer Name	Primer Sequences (5′-3′)	Primer Sequences
P21	F: 5′-3′ACCCCTTCCCCATACCC	XM_013977858.2
	R: 5′-3′TTCCTAACACCCATGAAACTG	
P27	F: 5′-3′GTCCCTTTCAGTGAGAACCG ATAC	NM_214316.1
	R: 5′-3′TTGCTGCCACATAACGGAATCAT	
PCNA	F: 5′-3′GTGATTCCACCACCATGTTC	NM_001291925.1
	R: 5′-3′TGAGACGACTCCATGCTCTG	
cyclin D1	F: 5′-3′GCGAGGAACAGAAGTGCG	XM_021082686.1
	R: 5′-3′TGGAGTTGTCGGTGTAGATGC	
cyclin E1	F: 5′-3′CTCGCCACTGCCTATACTGA	XM_005653265.2
	R: 5′-3′GGTGCCGCTGCATAAGGT	
CDK2	F: 5′-3′GCGAGGAACAGAAGTGCG	XM_013977858.2
	R: 5′-3′TGGAGTTGTCGGTGTAGATGC	
CDK4	F: 5′-3′GCGGAGATTGGTGTTGGTG	NM_001123097.1
	R: 5′-3′CATTGGGGACTCTTACGCTCTT	
Caspase3	F: 5′-3′TGCTGCAAATCTCAGGGAGACCT	NM_214131.1
	R: 5′-3′GTGCCTCGGCAGGCCTGAAT	
Caspase9	F: 5′-3′TGGCCTCGCTCTGGGATGCT	NM_02107526.7
	R: 5′-3′TGGCCTCGCTCTGGGATGCT	
Bcl-2	F: 5′-3′CTGCGAACCCGGTCTGCCTG	XM_005664627.3
	R: 5′-3′TCTCGGGCCCACTGCTCCTC	
Bax	F: 5′-3′CCGAGTGGCGGCCGAAATGT	XM_013998624.2
	R: 5′-3′TCCAGCCCAGCAGCCGATCTG	
GAPDH	F: 5′-3′GTGATTCCACCACCATGTTC	XM_021091114.1
	R: 5′-3′TGAGACGACTCCATGCTCTG	

## Data Availability

Not applicable.

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
