# Peer review of "Effect of Fumonisin B1 on Proliferation and Apoptosis of Intestinal Porcine Epithelial Cells"

_toxins, 2022, doi:10.3390/toxins14070471_

Round 1

Reviewer 1 Report

I carefully read the manuscript entitled ‘Effect of Fumonisin B1 on proliferation and apoptosis of intestinal porcine epithelial cells’. In my opinion, although the introduction is well written, there are some points about data presentation (results) to revise before possible consideration for publication on Toxins journal, as reported below:

-paragraph 2.1: add the percentage cell viability decrease after exposure to FB1 at 20 and 40 μg/mL for 48 h;

-Figure 1. Please revise the Figure legend. They are not clear. Specify the assay used in a, moreover, the data are obtained from how many replicates? Indicate also the statistical analysis used. In the legend of Figures 1 b and c add the microscope and the magnification. Moreover, in the corresponding Figures, the scale bars are not present or not clearly visible, please revise. Please, consider this comment also for other Figures and Figure legends;

-Figure 2a. the photos are not visible, please revise; change ‘µg/ml’ by ‘µg/mL’ in Figure 2b and check in overall text;

-The western blot analysis is not clear. I would like to see the overall western blot analysis. Moreover, why cleaved Caspase is placed upper? Add the molecular weight markers;

-paragraph 5.9: add the statistical test used;

-Revise the bibliography according to the journal guidelines.

Reviewer 2 Report

This manuscript is a continuation of the FB1 toxicity study that was recently published, probably by the same authors (Chen Z, Zhou L, Yuan Q, Chen H, Lei H, Su J. Effect of fumonisin B1 on oxidative stress and gene expression alteration of nutrient transporters in porcine intestinal cells. J Biochem Mol Toxicol. 2021 Apr;35(4):e22706. https://doi.org/10.1016/j.fct.2021.111977 Chen Z, Chen H, Li X, Yuan Q, Su J, Yang L, Ning L, Lei H. Fumonisin B1 damages the barrier functions of porcine intestinal  epithelial cells in vitro. J Biochem Mol Toxicol. 2019 Nov;33(11):e22397. https://doi.org/10.1002/jbt.22397).

I would like to give some recommendations to the authors of the manuscript:

Introduction: It would be useful to provide information here about the permissible concentrations of FB1 in corn or wheat as a raw material in order to compare them with the concentrations used in this study. It is necessary to somehow bring these characteristics into comparison in order to assess the real risk to animal health, for example. Otherwise, the choice of FB1 concentrations used by the authors in the study is not motivated.

Lines 57-59: It is necessary to rephrase the text (“In this study, we report our discovery that FB1 can stop the cell cycle and thus inhibit the proliferation of 58 IPEC-J2 cells by inhibiting the expression of a protein associated with cyclin. It is important to note that FB1 can also contribute to the mitochondrial apoptosis pathway of IPEC-J2 cells”), since the purpose of the study should be presented here, but not the main results of the study.

Line 65-66: End of phrase "...cell viability has decreased significantly” should be replaced with an exact percentage of reduced viability. In my opinion, the decrease was not “significant”, it was noticeable (about 10-12%). Am I right?

The quality of Figure 2a is unacceptable. There is no data, but someone can only see four black squares.

Line 184: There is the phrase “FB1 at 50 microns induced a large amount of DNA damage.” Please convert the concentration of 50 μM of FB1 to “μg/ mL” so that it can be compared the data with the results obtained in the manuscript.

Line 187: There is the following phrase “apoptosis was detected in mice fed 5 mg/kg FB1 for 42 days.” How can this information be correlated with the results obtained by the authors of the manuscript during the study?

Figure 4: Please change the title “Molecular mechanism of action of FB1 on IPEC-J2 cells" to ”The supposed mechanism of toxic action of FB1 on IPEC-J2 cells".

Line 202: Please change “with constant feed“ to “with continuous feed".

Line 244: Please change “polyvinylidene fluoride“ to "poly(vinylidene fluoride)”, since it is more correct.

All links must be given with all the necessary information! Now it is impossible to check all links due to the absence of the year of publication, the number of volume, pages, etc. Please see references 1 and 3.

Round 2

Reviewer 1 Report

The revised manuscript in my opinion can be accepted